# Adjuvant Effect of Molecular Iodine in Conventional Chemotherapy for Breast Cancer. Randomized Pilot Study

**DOI:** 10.3390/nu11071623

**Published:** 2019-07-17

**Authors:** Aura Moreno-Vega, Laura Vega-Riveroll, Tonatiuh Ayala, Guillermo Peralta, José Miguel Torres-Martel, Joel Rojas, Perla Mondragón, Adriana Domínguez, Rodrigo De Obaldía, Carlos Avecilla-Guerrero, Brenda Anguiano, Evangelina Delgado-González, Xóchitl Zambrano-Estrada, Olga Cuenca-Micó, Olivia De La Puente Flores, Alfredo Varela-Echavarría, Carmen Aceves

**Affiliations:** 1Instituto de Neurobiología UNAM-Juriquilla, Querétaro 76230, Mexico; 2Hospital General Regional #1 IMSS, Querétaro 76000, Mexico; 3Clínica Hospital Dr. Ismael Vázquez Ortiz ISSSTE, Querétaro 76000, Mexico

**Keywords:** molecular iodine, breast cancer, chemoresistance, immune response, transcriptomic analysis

## Abstract

This study analyzes an oral supplement of molecular iodine (I_2_), alone and in combination with the neoadjuvant therapy 5-fluorouracil/epirubicin/cyclophosphamide or taxotere/epirubicin (FEC/TE) in women with Early (stage II) and Advanced (stage III) breast cancer. In the Early group, 30 women were treated with I_2_ (5 mg/day) or placebo (colored water) for 7–35 days before surgery. For the Advanced group, 30 patients received I_2_ or placebo, along with FEC/TE treatment. After surgery, all patients received FEC/TE + I_2_ for 170 days. I_2_ supplementation showed a significant attenuation of the side effects and an absence of tumor chemoresistance. The control, I_2_, FEC/TE, and FEC/TE + I_2_ groups exhibited response rates of 0, 33%, 73%, and 100%, respectively, and a pathologic complete response of 18%, and 36% in the last two groups. Five-year disease-free survival rate was significantly higher in patients treated with the I_2_ supplement before and after surgery compared to those receiving the supplement only after surgery (82% versus 46%). I_2_-treated tumors exhibit less invasive potential, and significant increases in apoptosis, estrogen receptor expression, and immune cell infiltration. Transcriptomic analysis indicated activation of the antitumoral immune response. The results led us to register a phase III clinical trial to analyze chemotherapy + I_2_ treatment for advanced breast cancer.

## 1. Introduction

The main causes of the failure of breast cancer treatments are the rapid development of metastases and tumor resistance to antineoplastic drugs [1,2]. Anthracyclines (doxorubicin (DOX), epirubicin, etc.) are the gold standard in neoadjuvant therapy, and are commonly used in the 5-fluorouracil/epirubicin/cyclophosphamide or taxotere/epirubicin (FEC/TE) combination therapy for advanced breast cancer. However, even when treated with this potent chemotherapeutic combination, 30% of patients develop chemoresistance and cardiomyopathic side effects [3,4]. Therefore, continued research is necessary to develop novel breast cancer drugs that prevent these detrimental side effects.

In 1993, and later in 2004, two independent groups showed that molecular iodine (I_2_) supplementation at millimolar concentrations has beneficial effects in women with mammary fibrocystic disease or mastalgia, without any adverse effect on their thyroid function or general health [5,6]. Subsequently, several groups, including ours, have shown that these anti-proliferative and anti-inflammatory actions can also be observed in preclinical and clinical models of benign prostatic hyperplasia and cancer of various organs that uptake iodine [7,8,9]. We also reported that I_2_ exerts synergistic effects when used in combination with DOX antineoplastic treatments in both rodent [10] and canine [11] models. In our rodent model of methyl nitrosourea-induced mammary tumors, I_2_ treatment increased tumor sensitivity to DOX, allowing a four-fold reduction in the therapeutic dose of this drug. Moreover, I_2_ supplementation leads to significant cardioprotection, as indicated by a reduction in cardio lipoperoxidation and circulating cardiac creatine kinase (CK-MB) levels [10]. The molecular effects of I_2_ treatment also include decreased proliferation in DOX-resistant cells, decreased acquisition of epithelial-mesenchymal transition (EMT) markers, induction of tumor cell re-differentiation, and increased antitumor immune response [11,12]. In the canine study, iodine supplementation generated a significant attenuation of side effects, and disease-free survival increased by 33% compared with DOX- monotherapy studies after 10 months [11]. 

In the present study, we analyzed the clinical benefits as well as the molecular mechanisms of I_2_ using the following two protocols: (1) Protocol one utilized a daily supplement of I_2_ alone before surgery and, together with conventional FEC/TE therapy after tumor resection, in women diagnosed with stage II (Early) breast cancer; and (2) Protocol two, I_2_ was co-administrated with neoadjuvant FEC/TE therapy in women diagnosed with stage III (Advanced) breast cancer. Both treatments were followed up to determine the disease-free survival and over survival for 60 months. In the tumor samples, a transcriptomic analysis was carried out to describe the main molecular pathways involving the I_2_ response. 

## 2. Materials and Methods

### 2.1. Study Design

Patients diagnosed with breast cancer were eligible for the study. Other eligibility criteria included patients with normal hepatic, renal, and cardiovascular functions. Patients were excluded from the study if they were: 1) pregnant or breast-feeding or 2) diagnosed with any current or previous malignancy, with impaired bone, liver or kidney function or myocardial infarction during the last six months before the study, unstable or uncontrolled hypertension, or thyroid dysfunction. All subjects gave their informed consent for inclusion before they participated in the study. 

Since this a pilot study, the inclusion of a total of 40 patients was considered (10 patients per group). However, as it was predicted that 10–15% of the participants in each group could be lost or withdrawn from the study, 15 participants were initially assigned to each group. In the designation of each patient to the control or placebo groups, the dropper bottles of iodine and placebo were classified (numbered) and registered by a laboratory technician. In the patient recruitment, the dropper bottle assignment was sequential in number and not known by physicians or patients. The identification of the treatment that the patient received was known the day of the surgery by all the participants (doctors, researchers and patients). 

As described on the CONSORT flow diagram (Figure 1), two clinical study groups were established based on the stage of cancer diagnosed: Early (stage II) and Advanced (stage III) breast cancer group. In the Early group, 30 patients were randomly (double-blind) divided and received only molecular iodine (I_2_; 5 mg/day) or placebo (vegetable colored water) for 7–35 days (time designed by the oncologist’s protocol before surgery). After surgery, all patients received the I_2_ supplement, together with 4–6 cycles of a common chemotherapeutic cocktail containing FEC [5-fluoruracil (500 mg/m^2^), epirubicin (80 mg/m^2^), and cyclophosphamide (500 mg/m^2^)] or TE [Taxotere (docetaxel; 75 mg/m^2^) and epirubicin (80 mg/m^2^)]. In the Advanced group, 30 patients were randomly (double-blind) divided into the I_2_ or placebo groups, and both groups received neoadjuvant chemotherapy for 4–6 cycles of FEC of TE before surgery. I_2_ (5 mg/day) or a placebo were administered diluted in drinking water after breakfast, daily. After surgery, the two groups received the iodine supplement to complete 170 days, corresponding to the maximum period that some patients required to complete their chemotherapy treatment. In the Early group, the treatment was initiated on the day of enrollment of patients in the protocol, whereas in the Advanced group, the treatment began one week before the first chemotherapy cycle. At the end of adjuvant chemotherapy, all patients received radiation therapy and were treated simultaneously with tamoxifen or letrozole when hormone receptors were positive.

### 2.2. Adverse Reactions

Medical examinations as well as urinary, hematology and blood biochemistry analyses were performed twice (at the beginning of the protocol and on the day of surgery) in the Early group, and before surgery and before each chemotherapy cycle when the groups received FEC/TE. The treatment-related clinical adverse events were classified according The National Cancer Institute Common Toxicity Criteria V4.0 (CTCAE; [13]).

### 2.3. Evaluation of Treatment Efficacy

The tumor mass was measured at the beginning and at the day of surgery. The tumor response rate was graded according to the Response Evaluation Criteria in Solid Tumors (RECIST), which uses only the longest diameter of the tumors in the axial plane [14]. The response to each treatment was graded as progressive disease (PD) in tumors whose size increased by more than 25% compared with initial size, stable disease (SD) when no change in tumor growth was observed, and partial response (PR) when a decline in growth of more than or equal to 50% occurred; pathologic complete response (pCR) was defined as the disappearance of all lesions after four/six treatment cycles. 

### 2.4. Follow-Up

All patients were followed up for 20–60 months. Disease-free survival (DFS) was defined as the period that comprises from the first day after surgery to the date of recurrence or metastasis. Overall survival (OS) was defined as the period between the first day after surgery to the date of death or the last visit to the hospital reported.

### 2.5. Iodine and Thyroid Hormones

The total urine iodide level was measured with an ion chromatography system previously standardized in our laboratory [15]. Serum T_3_ and TSH levels were analyzed by an enzyme-linked immunosorbent assay (ELISA) (BioSource, Camarillo, CA, USA).

### 2.6. Molecular Markers Expression

The expression of *CDKN1A (P21*), *BAX, BCL2, BIRC5 (Survivin)*, and *HIF1A* was analyzed by quantitative real-time PCR (RT-qPCR) as described previously [16]. Total RNA was obtained using TRIZOL reagent (Life Technologies, Inc., Carlsbad, CA, USA), and the extracted RNA (2 μg) was reverse transcribed using oligo-deoxythymidine primers (RT). qPCR was performed with SYBRgreen dye in a Rotor-Gene 3000 detection system (Corbett Research, Mortlake, Australia) with the gene-specific primers listed in Appendix A. Gene expression was calculated by the 2^−ΔΔCt^ method and was normalized to the housekeeping gene *CTB* (β-actin). 

### 2.7. Immunohistochemistry

Immunochemistry analyses were carried out for ESR1 (ERα) and CD8 by peroxidase-labeled horseradish, and for CDH1 (E-cadherin) and Vimentin (VIM) by confocal microscopy. Cell death was analyzed by TUNEL (detailed in [17]).

### 2.8. Western Blotting

VEGF, PPARG, and actin protein expression levels were analyzed using a commercial kit (Roche Diagnostics GmbH, Mannheim, Germany) as described previously [16]. The membranes were treated with a polyclonal anti-VEGF antibody (SC-7269, diluted 1:1000; secondary antibody-SC-2060; diluted 1:500), a polyclonal anti-PPARG (SC7196, diluted 1:1000; secondary antibody SC-2004, diluted 1:500) and a polyclonal anti-actin antibody (SC-1616, diluted 1:10,000; secondary antibody SC-2020, diluted 1:1000) (all antibodies were from Santa Cruz Biotechnology Inc, Dallas, TX, USA). Protein bands were visualized by chemiluminescent detection (ECL, Amersham Biosciences, Buckinghamshire, UK) and densitometry was performed with Image Lab™ software (Bio-Rad Laboratories, Los Angeles, CA, USA).

### 2.9. RNA-Seq and Transcriptomic Analysis

Detailed constructions and all specific data analyses including pathway and upstream regulator prediction, as well as all other analyses using the transcriptomic data, can be found in protocols.io [17]. The full annotated sequences from the RNQ-sequencing are available at the European Nucleotides Archives web site (https://www.ebi.ac.uk/ena/erp110028).

### 2.10. Statistics Analysis

The effects of treatment on tumor response, clinical response and toxicity were analyzed using 2 × 2 contingency tables and chi-square tests. The impact of treatment on clinical response and the other variables were analyzed with one-way ANOVA and Tukey´s honest significant difference tests. Non-parametric data was analyzed by the Mann-Whitney *U* test. Survival data were described and compared with the Kaplan–Meier estimator. *P*-values less than 0.05 were considered statistically significant.

### 2.11. Study Approval

The local medical and scientific bioethics committee at all three participating centers approved the study protocol (INB-UNAM-004.H; IMSS-HGR1: 185-09-03-05/MPSS, ISSSTE: 22-205/CEI 248/2009) and it is registered at Clinicaltrial.gov (NCT03688958). The protocols used in this study also conform to the principles of the Declaration of Helsinki. All patients provided written informed consent before study enrollment.

## 3. Results

### 3.1. Patients

Between January 2006 and September 2008, a total of 30 patients diagnosed with Early stage breast cancer were enrolled in the study and 23 completed the treatment (Appendix A). The average age of the study group was 55.2 ± 12.7 years with an interval of 38–81 years. Tumor classification showed that 83.3% of the patients had ductal-, 10% had lobular-, and 6.7% had medullar-type tumors. Furthermore, between January 2009 and May 2010, a total of 30 patients diagnosed with Advanced stage breast cancer were enrolled in this study and 22 completed the treatment (Appendix A). The average age of this group was 45.0 ± 9.7 years with an interval of 29–67 years. Tumor classification showed that 80% had ductal- and 20% had lobular-type tumors.

### 3.2. Toxicity and Safety

Patients in the Early group (Placebo/Control or I_2_ treatment) did not report any discomfort or iodine toxicity during treatment before surgery. In the chemotherapy groups (FEC/TE and FEC/TE + I_2_), we used The National Cancer Institute Common Toxicity Criteria V4.0 (CTCAE; [13]) to evaluate toxicity. As shown on Table 1, the most common hematologic toxicity was neutropenia in grades 1–3 and was presented 8 days after the second or third treatment cycle, the presence of I_2_ supplementation prevented the severity (41.6 to 63.6% of the patients in grade 1), since grades 2–3 were present only in 4 patients (36.4%) who received the FEC/TE treatment alone. In most patients, the hematological restoration occurred 15 days after chemotherapy. Neither febrile aplasia nor thrombocytopenia were observed in any of the patients. Gastrointestinal discomfort (nausea, vomiting and diarrhea) from grade 1 or 2–3 was observed in all patients in the FEC/TE group (9 and 2 patients respectively), while only 1 to 3 patients in the FEC/TE + I_2_ groups presented discomfort in grade 1. Moreover, 70–80% of patients reported moderate muscle pain (myalgia), but only 3 patients (27.3%) in the FEC/TE group developed dermatological toxicity (hand-foot syndrome). No metabolic or biological toxicity was reported in any patient.

### 3.3. Tumoral Response

The residual tumor size of each tumor, as well the tumoral response classification (RECIST; [14]), are summarized in Figure 2. In patients with early breast cancer that received only placebo before surgery (control group), five patients (45.5%) exhibited progressive disease and six patients (54.5%) maintained a stable tumor size (stable disease). Regarding the group with I_2_ supplement alone, in eight patients (66.7%) tumor growth ceased (SD), and the size of 4 tumors (33.3%) diminished, resulting in a partial response. Several tumors from advanced breast cancer patients treated with neoadjuvant FEC/TE showed a significant response, leading to PR in 6 patients (54.6%), and pathologic complete response (pCR) in 2 patients (18.1%). However, in 3 patients (27.3%), tumors became chemoresistant, exhibiting progressive disease (one maintained its size, and the other two continued their growth). The FEC/TE + I_2_ group obtained the best responses: with 7 patients (63.7%) having shown PD, 4 patients (36.3%) obtained pathologic complete response (pCR) and chemoresistance was absent (0%). This was the only group with an objective response rate (ORR) of 100%.

### 3.4. Follow Up

The recurrence and survival results of the five-year follow up are summarized in Figure 3. Three patients (27%) in the control group, two patients (17%) in the I_2_ group, 6 patients (54%) in the FEC/TE group and 2 patients (18%) in the FEC/TE + I_2_ group relapsed or experienced metastasis with a mean disease-free survival rate of 35 ± 13, 45 ± 7, 29 ± 6 and 49 ± 1.4 months (Figure 3A). Regarding OS, the number of deceased patients was two patients in the control (18%) and one patient (0.8%) in the I_2_ supplemented early groups, and four patients (36%) and one patient (9%) in FEC/TE and FECT/TE + I_2_ advanced groups respectively. The Kaplan-Meier analysis (Figure 3B,C) indicated that there is only one statistically significant decrease in DFS for the FEC/TE treated group when compared to the rest of the other treatment groups (*P* = 0.04). For this same treatment group (FEC/TE), we could only observe a tendency of decrease of OS when compared to the other treatment groups (*P* = 0.08, ns).

### 3.5. Molecular Tumoral Analysis

At the molecular level, I_2_ treatment arrested cell cycle, induced apoptosis (increased expression of *CDKN1A*, increased DNA fragmentation-TUNEL, and increased *BAX/BCL2* index as well as decreased *BIRC5* expression) and impaired the expression of vascularization factors (decreased hypoxia-inducible factor (HIF1A) and vascular endothelial growth factor (VEGF) (Figure 4A,B). All these changes might be associated with the observed increase in PPARG protein expression (Figure 4C) as we have observed in our previous studies [10,11]. Notably, the biopsy (pre-treatment) and tumor samples (after treatment) obtained from all patients diagnosed with stage II were estrogen receptor ESR1-positive regardless of I_2_ supplementation (Figure 4D). In patients diagnosed with stage III breast cancer, 45% of biopsies (5 patients per group) were negative for ESR1 expression. After treatment, the tumor from one patient in the FEC/TE group became ESR1 positive, whereas in the FEC/TE + I_2_ group, 90% of the tumors (10 samples) were ESR1 positive. This suggests that I_2_ supplementation favors the re-induction of ESR1 expression in breast cancer.

The urine iodine level reflected I_2_ ingestion, and the steady circulating levels of triiodothyronine (T_3_) and thyroid-stimulating hormone (TSH) indicated that I_2_ treatment did not alter thyroid status (Appendix A). A moderate but significant increase in creatine kinase-MB (CK-MB) level was observed in the FEC/TE patients, implying cardiotoxicity (Appendix A). In contrast, in the I_2_-treated FEC/TE patients, the serum level of this enzyme remained equivalent to those of the control groups, indicating a preventive effect of treatment. 

### 3.6. Effect of I_2_ and/or FEC/TE + I_2_ Treatment on General Tumor Gene Expression

Common genes showing statistically significant changes in expression levels (Differentially Expressed Genes; FDR < 0.05; |log fold change (FC)| > 1) in all treatments (compared with those of the control) were identified. Figure 5A shows the unsupervised clustering of each treatment based on the expression of these common differentially expressed genes (DEGs). Through functional enrichment analysis [18], we identified the top ten significantly enriched pathways related to I_2_ or FEC/TE+I_2_ treatment (Figure 5B; Intersection of Venn Diagrams, genes with FDR ≤ 0.05; |logFC| > 1). In the I_2_ treated group, the most significantly enriched pathways associated with upregulated genes were related to cell differentiation, cell adhesion molecules (CAM), and inflammatory/immune responses, whereas the pathways associated with the downregulated genes (compared with control group) were related to invasive potential (extracellular matrix-receptor interaction), survival/antiapoptotic pathways (PI3K-AKT), and chemoresistance (drug metabolism). Interestingly, the enriched pathways in patients with early breast cancer treated with I_2_ alone and those in patients with advanced breast cancer treated with FEC/TE + I_2_ were practically the same, suggesting common I_2_ effects (Figure 5B). 

### 3.7. Analysis of EMT-Related Gene and Protein Expression after I_2_ Treatment

The analysis of our transcriptomic data focusing on an EMT gene signature derived from MSigDB [19] showed that I_2_ supplementation alone or in coadjuvancy with FEC/TE downregulates the expression of EMT-related genes such as *VCAM1*, *PTHLH*, *LAMA1*, *MMP3*, *BDNF*, and *TNC* when compared with that of the untreated controls (Appendix A). Furthermore, the immunohistochemical analysis of individual tumor samples showed that I_2_ treatment in patients with early breast cancer and in co-adjuvant (FEC/TE + I_2_) treatment in patients with advanced tumors tended to increase the expression of membrane localized E-cadherin (CDH1) (Appendix A). In contrast, vimentin (VIM) expression in all treatment groups was lower and the same trends were observed in cells double stained with both CDH1 and VIM (Appendix A), indicating an ongoing EMT process.

### 3.8. Immune Response Activation after I_2_ Treatment

The results of the analysis of the predicted immune response following treatment are summarized in Figure 6A. We first predicted the abundance of infiltrating immune and stromal cell populations using our transcriptomic data and the MCP counter R package [20]. We observed that after I_2_ treatment, the number of CD8^+^ T cells, dendritic cells, cytotoxic lymphocytes, and B-lineage cells appears to be enriched when compared with the control group (and for some cell types compared also with the FEC/TE group). These results suggested that I_2_ supplementation increases tumor immune infiltration and that the presence of FEC/TE may synergize such an immune response. To confirm these observations, immunohistochemical analysis was carried out on individual tumor samples. 

As shown in Figure 6B (H&E), the tumors from patients who received I_2_ or FEC/TE + I_2_ treatment presented a statistically significant increased number of infiltrating lymphocytes compared with the control groups. The increase in the number of CD8^+^ lymphocytes supports the activation of an antitumoral immune response. Moreover, tumors from FEC/TE + I_2_-treated patients had multiple necrotic areas (data not shown). We also observed a significant reduction in the epithelium compared with the stromal content in the FEC/TE + I_2_ group (Figure 6B, Masson’s trichrome).

Based on our MCP counter enrichment analysis, we further evaluated CD4^+^ T cell differentiation in order to determine whether an immune infiltration exerted an antitumor response. We determined this by focusing on the differences in the expression of genes associated with naive CD4^+^ T cell Th1 or Th2 differentiation (Appendix A). 

These data show that the logFC (FDR ≤ 0.05) values of various genes are significantly different between the treatment groups. Furthermore, it appears that I_2_ and FEC/TE + I_2_ treatments favored the overexpression of genes related to Th1 differentiation compared with that of the untreated control. Furthermore, we analyzed if the T CD8^+^ infiltrated cells could have effector or exhausted properties. We selected five T CD8^+^ effector markers (perforin, granzymes A and B, Tbx21 and EOMES) and two markers for exhausted T CD8^+^ cells (PD-1 and CTLA-4) and examined their expression in our transcriptomic data. All genes were upregulated in groups with I_2_ treatment compared to the control groups (data not shown). Altogether, Th1 polarization and the presence of T CD8^+^ effector markers seem to support an antitumor immune response.

### 3.9. Transcriptional Regulation of Common Genes in the I_2_- and FEC/TE+I_2_-Treated Tumors

Further exploration of the networks involving the common DEGs in I_2_- and FEC/TE+I_2_-treated tumors enabled us to predict their main upstream transcriptional regulators [21]. For genes that were significantly upregulated following the I_2_ treatment, we found five transcription factors (TFs), IRF4, STAT1, SPI1, HIVEP1, and E2F1, that could regulate 76% of these genes (Figure 7). In addition, we identified IRF1 as a key upstream regulator of all these TFs. We could not identify regulatory factors for 24% of the common upregulated genes in the I_2_-treated tumors; however, a closer look at these genes revealed that many of them are pseudogenes, long non-coding RNAs, or immunoglobulins. These findings indicate that most of the coding genes are regulated by the TF network outlined in Figure 7A. With respect to the common downregulated genes in the I_2_-treated tumors, their expression could be largely controlled by PARP1, DAZAP1, BARX1, SUZ12, and MTF1, with PARP1 being the key regulator in this network (Figure 7B). Notably, another group of the downregulated genes, which included cytochromes, appeared to be involved in drug metabolism.

### 3.10. Role of PPARG in the Observed I_2_-Mediated Changes

Having previously demonstrated that the activation of PPARG is via the formation of 6-IL [16] and observing that PPARG protein expression appears to be increased in the I_2_-treated patients in our study, we explored the possible role of these receptors in the observed I_2_-mediated changes in gene expression and regulation. Following experimentally supported datamining [22,23], we found an established feedback loop between PPARG and IRF1 (Figure 8A) as well as possible interactions between this receptor and other main TFs identified. Interestingly, we found similar TF expression changes in IRF1, STAT1, and IRF4 in the I_2_- and FEC/TE+I_2_-treated groups. To further explore the possible role of PPARG in this network, we analyzed the expression of genes positively correlated with PPARG in each of our experimental groups [METABRIC data [24], Pearson correlation ≥ 3; Figure 8B). Appendix A summarizes the ontology enrichment of the most significant biological processes (FDR < 0.05) associated with the different upregulated or downregulated genes in each experimental dataset. It appears that the PPARG correlated genes that were upregulated in patients with early breast cancer treated with I_2_ and in those with advanced breast cancer treated with FEC/TE + I_2_ are associated with an immune response, whereas those that were downregulated appeared to be related to cell motility, when compared with those in the control groups. 

## 4. Discussion

Breast cancer in women is a serious health concern globally. However, several treatments result in chemoresistance or are not effective enough to stop tumor growth. This is a pilot study aimed to evaluate the adjuvant effect of I_2_ supplementation in two patient cohorts treated with conventional chemotherapy for stage II and stage III mammary cancer. The FEC/TE cocktail is the standardized treatment used by the national public health services in Mexico, and the assignment of patients to each therapy group in this study was done by the oncology team. Our data corroborated those of previous reports showing the powerful antineoplastic effects of FEC/TE, as well as the induction of significant side effects [3,4]. One of the most interesting results obtained in the present study is the clear attenuation of severe side effects, such as diarrhea, neutropenia, and neuropathy, as well as the complete prevention of hand-foot syndrome and cardiotoxicity when I_2_ is supplemented during FEC/TE treatment. These results agree with those of our previous studies in rat and canine models, in which I_2_ supplementation mediated similar health benefits and cardioprotection [10,11]. Several studies have shown that FEC/TE-induced cardiotoxicity, specifically that mediated by anthracyclines, is caused by the generation of reactive oxygen species (ROS) [4]. Therefore, the specific cardioprotective function (and potentially the other noted benefits) of I_2_ treatment may be explained by the well-established antioxidant effects of this form of iodine, which is ten times more efficient than ascorbic acid and 100 times more potent than KI in vitro [10]. Indeed, it has also been reported that cardiomyocytes under ischemia-like conditions take up iodine and produce protective thyroid hormones [25]. Moreover, these antioxidant effects have also been associated with the attenuation of inflammation and pain symptoms in patients with fibrocystic breasts [5,6], as well as in a testosterone-induced oxidative stress model of prostatic hyperplasia in rats when treated with the same concentrations of I_2_ [26].

Another notable result observed in the present study is the clear antineoplastic effect exerted by the I_2_ supplement independently of cancer stage. In both conditions, the presence of iodine arrests tumor growth and decreases the expression of genes involved in invasion and chemoresistance. Moreover, the supplementation with I_2_ seemed to potentiate the cytotoxic effects of FEC/TE, as seen by the doubling of the number of patients with complete response and by the total inhibition of chemoresistance compared to that observed in patients treated with FEC/TE alone. In a previous study [27] that used neoadjuvant treatments with FEC/TE in advanced mammary cancer, the DFS and OS were 46.9% and 59.4% and the mean survival time of 52.4 ± 18 months. In the present pilot study, the I_2_ supplement before and after surgery (FECT/TE + I_2_ group) significantly increased the number of patients without relapse or metastasis in comparison with the group that received I_2_ supplement only after surgery (FEC/TE group) (83% versus 46%) with a mean of 49 ± 1.0 versus 34 ± 14 months (*P* = 0.04). This trend was also observed in the 5-year OS, showing only one death (9%) in the FECT/TE + I_2_ group whereas 4 patients deceased (36%) in the FEC/TE group. Our study did not show statistically significant differences in the analysis of overall survival (*P* = 0.08), nonetheless the survival time was longer in patients who received I_2_ throughout the treatment (before and after surgery) regardless of the stage of cancer (50 versus 35 ± 4.0 months). 

To understand the mechanisms underlying the potent action of I_2_, we used a variety of molecular techniques. Our results strongly suggest that I_2_ alone induces apoptosis and tumoral re-differentiation, as indicated by increased TUNEL staining, and membrane re-localization of E-cad (CDH1) as well as expression of ESR1, respectively. In combination with FEC/TE, I_2_ treatment exhibited a synergic effect on cancer cell apoptosis and impaired expression of chemoresistance markers (*BIRC5* and *HIF1A*) as well as changes in the expression of specific EMT markers (increased E-cad expression and decreased Vimentin expression). This I_2_-mediated response, which was observed in individual tumor samples, also agreed with our transcriptomic analysis results in which the primarily upregulated pathways corresponded to cell differentiation and cell adhesion molecules (CAM), whereas the downregulated pathways corresponded to invasive potential (ECM-receptor interaction), survival/antiapoptotic pathways (P13K-AKT), and chemoresistance (drug metabolism). Several groups have proposed various mechanisms and pathways that mediate the cellular actions of I_2_. For example, oxidized iodine may directly induce apoptosis by dissipating the mitochondrial membrane potential [9,28] or can function indirectly through the formation of iodolipid and induction of re-differentiation via PPARG activation [16]. In the present study, *PPARG* mRNA expression was unchanged, whereas its protein expression increased following I_2_ treatment, suggesting a post-translational regulation of this receptor. PPARG was initially described as a TF involved in adipose tissue differentiation, and its synthetic agonists, including thiazolidinedione drugs, are used for the treatment of type II diabetes [29]. In cancer therapy, several studies have shown that PPARG activation exerts antiproliferative, apoptotic, differentiation, and antiangiogenic effects [30]. In fact, agonists of these receptors are currently being tested in clinical trials as antineoplastic alternatives. However, the results reported to date are inconclusive, ranging from a lack of effect to a moderately helpful response when used as a monotherapy [31]. Interestingly, when some of these agonists were used in combination with conventional antineoplastic drugs, such as carboplatin or tumor necrosis factor-related apoptosis-inducing ligand (TRAIL), synergistic effects were observed, indicating that differentiation induced by PPARG activation restores sensitivity to the cytotoxic drugs [32,33,34]. Such re-differentiation effects were also apparent in our previous preclinical studies using I_2_ in combination with DOX [10,11] as well as in a DOX-resistant cell model [12], supporting the notion that PPARG activation mediates some of the effects of I_2_ treatment.

We also noted a significant anti-tumoral immune response following I_2_ treatment. The pathway enrichment analysis of our transcriptomic data show that I_2_ treatments are accompanied by an increase in the number of CD8^+^ T, dendritic, and B-lineage cells. Moreover, 76% of the upregulated genes can be modulated by IRF1. Originally identified as a transcriptional regulator of interferon genes, IRF1 also regulates several target genes that play essential roles in the physiological and pathological processes, including several aspects of the innate and adaptive immune responses [35]. In the context of cancer, IRF1 is required for Th1 polarization in NK cells, mature CD8^+^ T cells, and M1 macrophages. In addition, IRF1 is involved in the negative regulation of Treg cells leading to the reactivation of an anti-tumor immune response [36]. Our analysis also shows that although IRF1 can stimulate genes related to both Th1 (*STAT4*, *T-bet/TBX21*, *IL12RB1*, and *IL12RB2*) and Th2 (*IRF4*, *E2F1*, and *FOXP3*) responses, the predominant activation type in both I_2_-treated groups seemed to be Th1. Further, in both the early I_2_- and advanced FEC/TE+I_2_-treated groups, we found a significant upregulation of Th1 differentiation-associated genes (*T-bet /TBX21*, *IL12RB1*, *IL12RB2*, *STAT1*, *STAT4*, *TNF*, and LTA) and a significant downregulation of some Th2 differentiation-associated genes (*GATA3*, *TSLP*, and BHLHE41). In agreement with our transcriptomic results, our immunohistochemical analysis results showed significant infiltration of CD8^+^ lymphocytes in the I_2_-treated patients. Moreover, in the FEC/TE+I_2_-treated tumors, lymphocyte infiltration was also accompanied by an increase in necrotic areas (data not shown), indicating an important anti-tumoral effect. Similar results were also observed previously in our in vivo canine and mice xenograft cancer models [11,37]. 

Notably, molecular iodine has been proposed as an immune modulator [38,39]. It has been demonstrated that several immune cell types can internalize I_2_, and depending on the cellular context, this element can act as an anti- or pro-inflammatory agent [40]. The anti-inflammatory effects of I_2_ are modulated by the suppression of production of reactive oxygen intermediates in polymorphonuclear cells [41] or by the inhibition of neutrophil chemotaxis [42]. In contrast, I_2_ could also potentially act directly on immune cells and induce their reactivation. For example, in chronic wounds, I_2_ activates the influx of macrophages and T cells [43]. In vitro, I_2_ has also been shown to enhance TNFα secretion from macrophages stimulated with bacterial lipopolysaccharides [43,44] and induce the release of anti-tumoral cytokines, such as IL-6, IL-10, and IL-8-CXCL8, in leucocytes [39]. Furthermore, although the complete effects of this halogen on the immune response have not been evaluated, an analysis of the expression of *PPARG*-positively-correlated genes [24] in our dataset suggests that in the I_2_-treated groups, *PPARG* is only weakly activated. However, the PPARG coactivator 1 Beta (*PPARGC1B*) was positively induced after I_2_ treatment (data not shown) and this increase could be correlated with the observed increase in PPARG protein concentration, suggesting that the effects of I_2_ are mediated by a complex regulatory pathway.

As a pilot study, our findings should be interpreted within the context of certain limitations. The number of patients was not sufficient to identify a statistically significant effect of iodine supplementation on disease-free survival in women with early cancer, and for both groups in the overall survival rate. Also, although most tumors, regardless of stage, can be considered luminal, the lack of Her2 marker identification limits the analysis of this factor in the response of the iodine supplement. However, the consistent attenuation of side effects and the antineoplastic response (less invasion, inhibition of chemoresistance, and activation of the immune system) associated with the presence of this halogen, highlight the relevance of a phase III study to analyze the supplement of I_2_ in conventional treatments against advanced breast cancer.

## 5. Conclusions

This is a pilot study that analyzed the adjuvant effect of molecular iodine together with the most widely used chemotherapeutic treatment in Mexico against breast cancer. We evaluated the effects of I_2_ during the initial and advanced stages of breast cancer with respect to toxicity, tumor response, survival at 5 years, and transcriptomic response. Our data indicate that supplementation with I_2_ improves the effectiveness of the treatment, decreasing side effects and increasing disease-free survival specially in advanced conditions (stage III). We also show that iodine supplementation induces tumor re-differentiation and the reactivation of antitumor immune responses. This study establishes a framework for the proposal of a phase III study for the analysis of iodine supplementation in the treatment of advanced breast cancer.

## 6. Patents

Aceves C, et al. IMPI. Mx/E/2012/084405. México DF. 14/Nov/2012, PCT/Mx2013/000139. MX/E/2017/009914. April, 1919 2017. Validity: 14/11/2012-14/11/20132.

## Figures and Tables

**Figure 1 nutrients-11-01623-f001:**
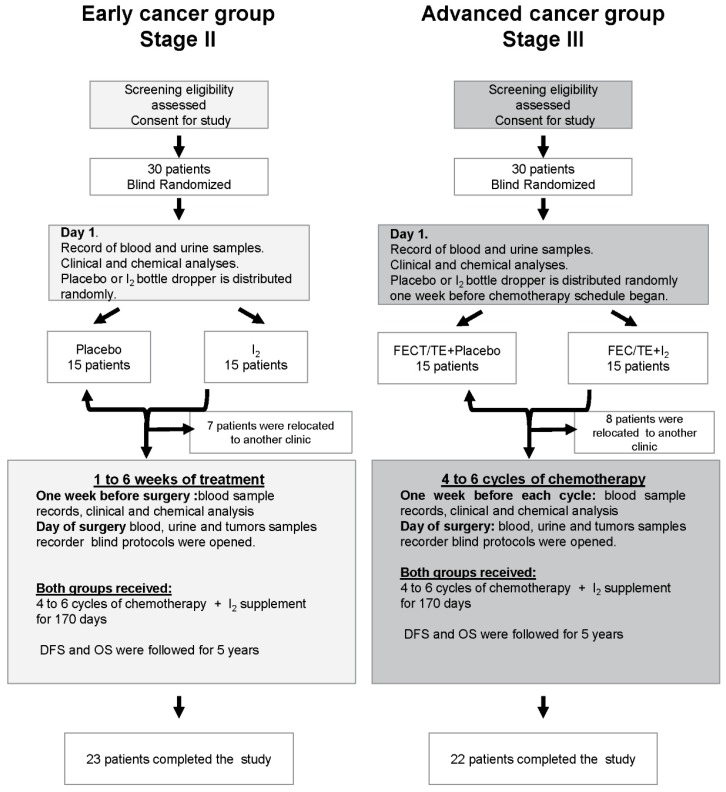
Flow diagram of study design.

**Figure 2 nutrients-11-01623-f002:**
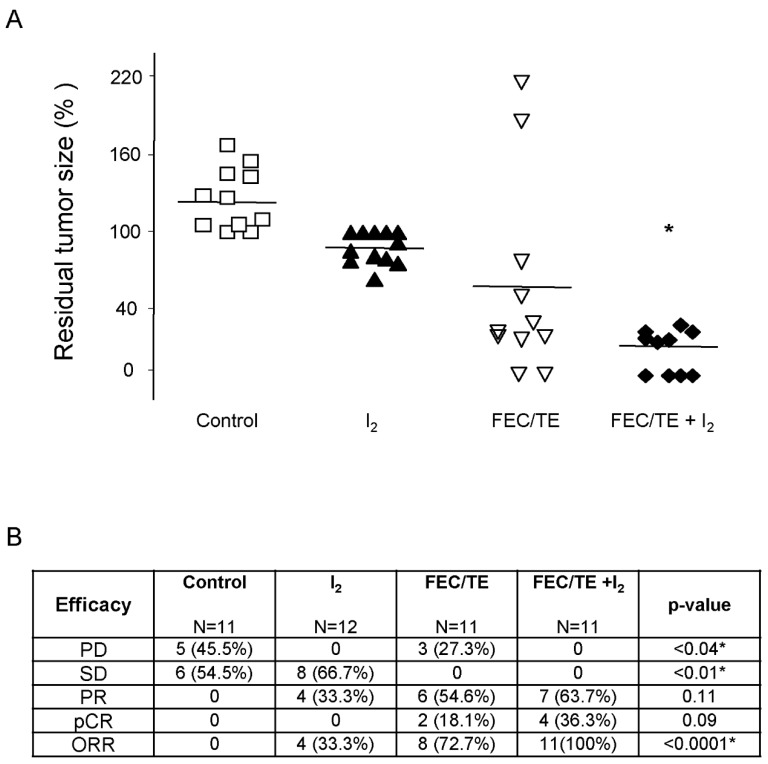
Tumoral response. (**A**) Changes in tumor size (%) between the first diagnostic (mammography) and final tumor size (surgery day). Dots represent individual patients. * *P* < 0.05, between the FEC/TE + I_2_ group and all other groups. (**B**) Response evaluation criteria in solid tumors (RECIST); PD, progressive disease; SD, stable disease; PR, partial response; pCR, pathologic complete response. ORR, objective response rate. *P*-value by chi-square test.

**Figure 3 nutrients-11-01623-f003:**
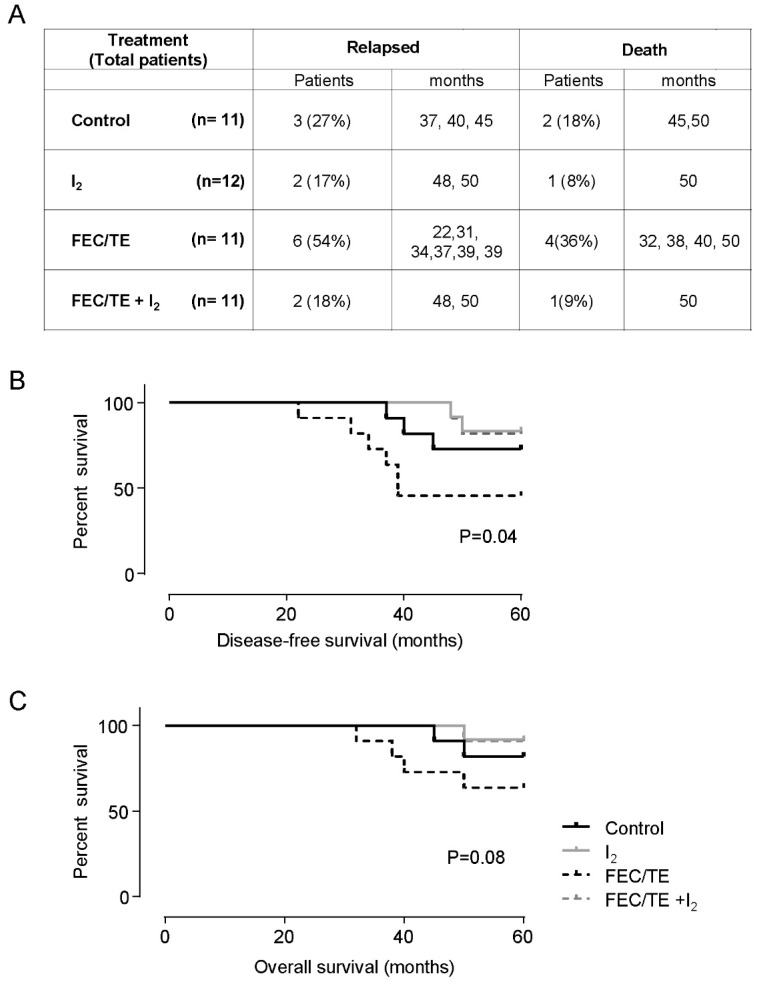
Recurrence and survival rate. (**A**) Patients who relapsed or died; number of patients and the months of relapse or death. (**B**) Analysis of disease-free curve; and (**C**) analysis of the overall survival curve. Kaplan-Meier analysis.

**Figure 4 nutrients-11-01623-f004:**
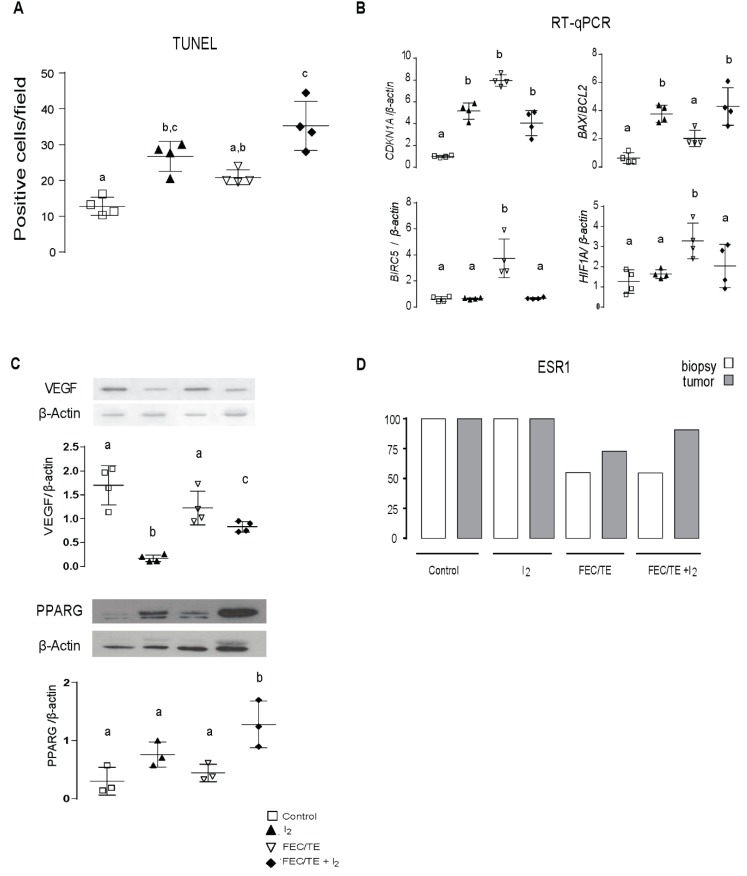
Molecular response to iodine, FEC/TE, or both (FEC/TE+I_2_) in tumor samples. (**A**) Quantitative results of the TUNEL assay. (**B**) RT-qPCR analysis of cell cycle arrest (*CDKN1A*), apoptosis index (*BAX/BCL2*), anti-apoptosis (*BIRC5*-Survivin), and hypoxia (*HIF1A*). Each dot represents an individual sample. (**C**) Western blotting analysis of vascular endothelial growth factor (VEGF) and peroxisome proliferator-activated receptor type gamma (PPARG) protein expression. (**D**) Percentage of positive patients with estrogen receptor alpha (ESR1) expression in biopsy and tumor samples. Gene expression was calculated by the 2^−ΔΔCt^ method using β-actin as a housekeeping gene. Data are expressed as mean ± SD. Each dot represents an individual sample, and the different letters or asterisk indicate significant differences between groups (*P* < 0.05).

**Figure 5 nutrients-11-01623-f005:**
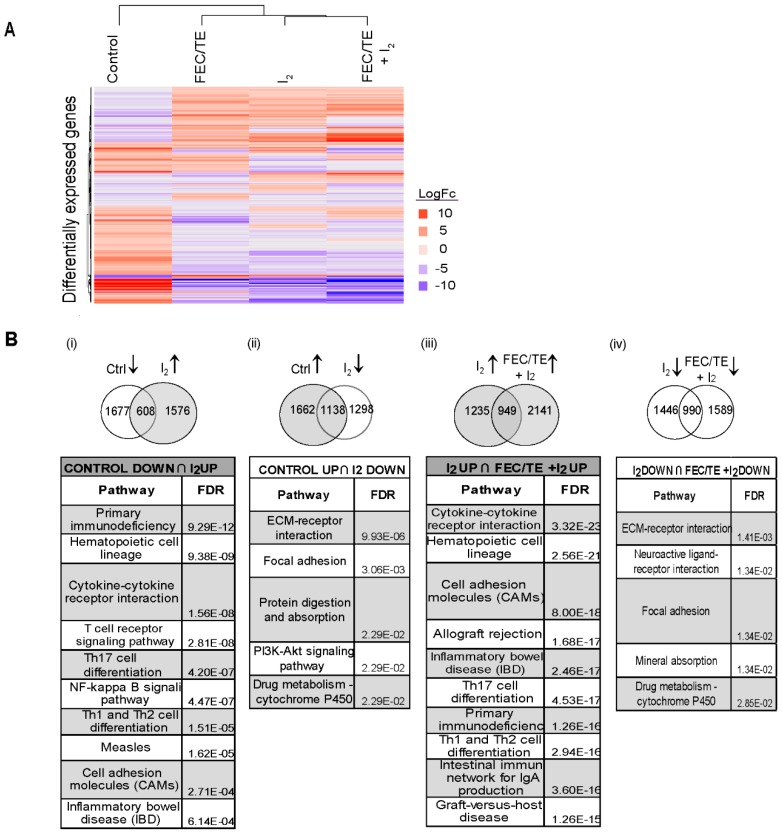
Effects of I_2_ treatment on gene expression. (**A**) Hierarchical clustering of different treatment groups based on gene expression. Analyses were performed with 855 (4%) common genes that exhibited a significant change in expression in any of the treatment groups (FDR < 0.05, |log fold change (FC)| > 1). (**B**) Pathway enrichment analysis of I_2_ supplementation. The Venn diagrams between the experimental groups represent the intersections of interest to analyze for pathway enrichment: (i) and (ii) genes “normalized” by I_2_ supplementation and (iii) and (iv) common genes influenced by I_2_ supplementation in both early and late stage breast cancer. Arrows indicate upregulation or downregulation. The bottom panels show the top enriched KEGG pathways in each group. All pathways shown have an FDR (B&H) < 0.05.

**Figure 6 nutrients-11-01623-f006:**
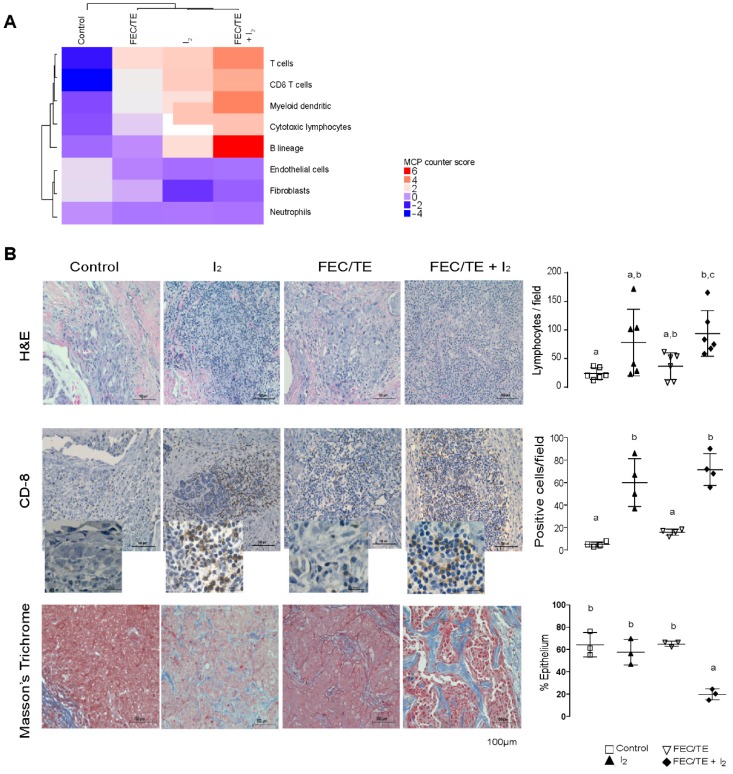
Immune signature and differential response of epithelial and lymphocytic infiltration after I_2_ treatment. (**A**) Abundance of different cell populations was calculated by the MCP-counter quantification method, based on a gene expression matrix of the distinct experimental groups. Hierarchical clustering of experimental groups. (*n =* pool of four individual tumors per experimental group. Duplicated pools per group were used for gene expression calculation). (**B**) Representative micrograph images of H&E and CD8 immunohistochemistry as well as Masson’s trichromic staining. The graphs on the right show the number of lymphocytes, CD8+ lymphocytes, and Masson’s trichromic positive cells per field. These analyses were performed as the average of three random fields (40×), and quantification was performed using ImageJ 1.47. Data are expressed as mean ± SD. Each dot represents an individual sample, and different letters indicate significant differences between groups (*P* < 0.05).

**Figure 7 nutrients-11-01623-f007:**
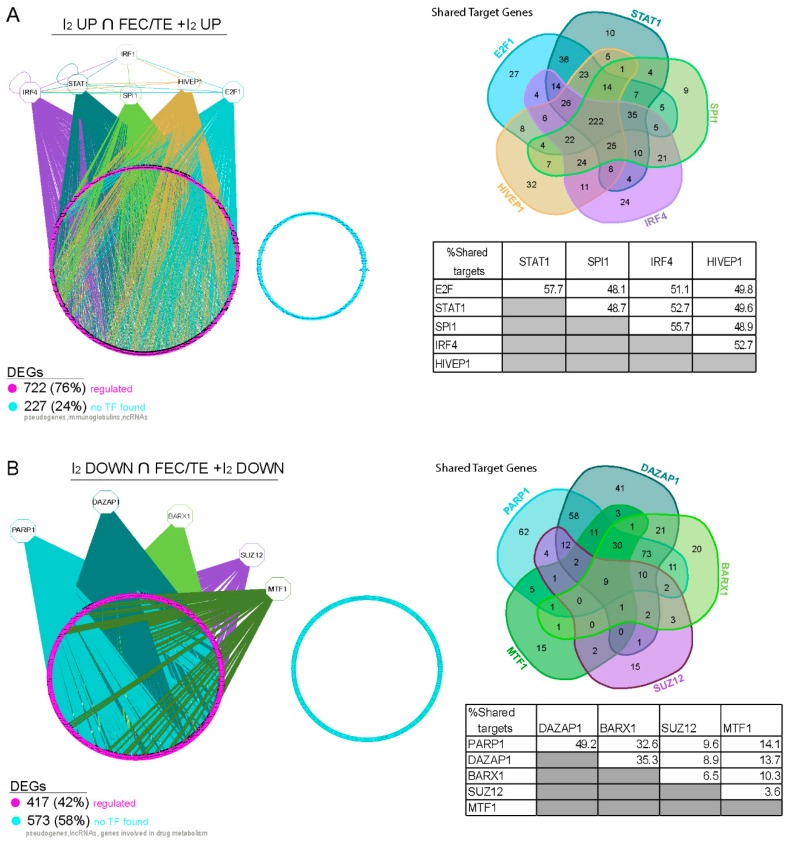
Master regulators involved in I_2_ treatment-related gene regulatory networks. (**A**) The main candidate transcription factors predicted to regulate genes found upregulated following I_2_ or FEC/TE + I_2_ treatment. (**B**) The main candidate transcription factors predicted to regulate genes found downregulated following I_2_ or FEC/TE + I_2_ treatment. For both panels: Left, iRegulon predicted gene regulatory network. The predicted target genes are shown in pink, whereas genes for which no transcription factor regulator was identified are shown in blue. Right, Venn diagram and diagram of shared target genes between the main transcription factors identified. The identified transcription factors have a maximum FDR on motif similarity of 0.001. Differentially expressed genes (DEGs) are defined as having an FDR < 0.05 and |log fold change (FC)| > 1).

**Figure 8 nutrients-11-01623-f008:**
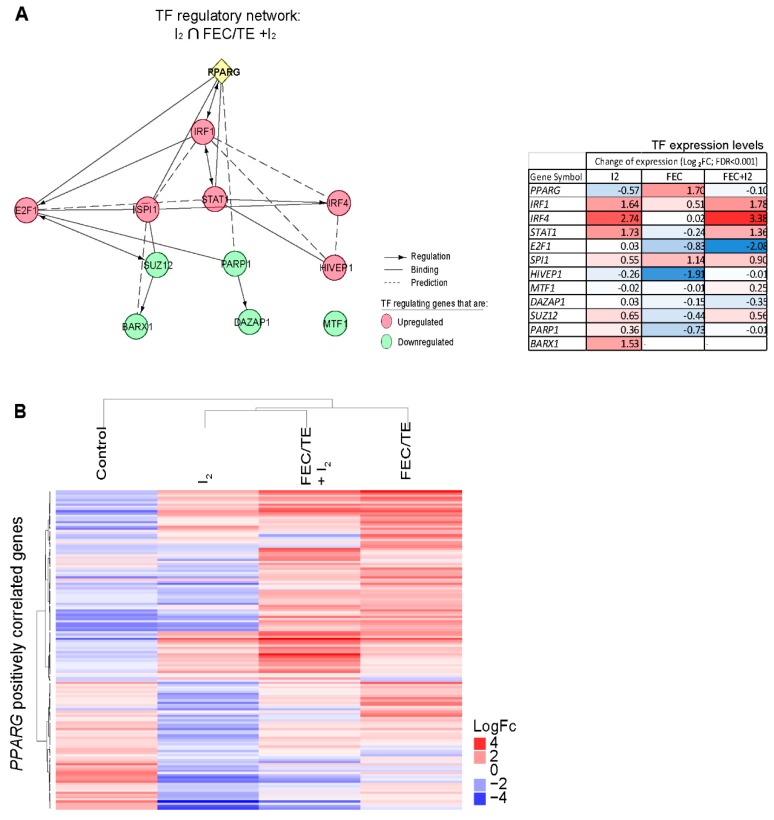
Possible role of PPARG in the identified I_2_ treatment-related regulatory networks in breast cancer. (**A**) Possible interaction networks involving PPARG, and the main transcription factor networks regulating the I_2_-modulated genes in our breast cancer model. The network was constructed using Reactome and text-mining databases. The table on the right shows the gene expression changes (log_2_FoldChange) in network components in different treatment groups. (**B**) Non-supervised hierarchical clustering of treatment groups based on genes that positively correlated with PPARG (Pearson ≥ 0.3) from the Breast Cancer METABRIC study. All genes had an FDR ≤ 0.05. *n =* pool of four individual tumors per experimental group. Duplicated pools per group were used for gene expression calculation.

**Table 1 nutrients-11-01623-t001:** Treatment-related clinical adverse events.

Adverse Events	Control/Placebo + FEC/TE + I_2_ *N* = 11	I_2_ + FEC/TE + I_2_*N* = 12	FEC/TE + Placebo+FEC/TE + I_2_ *N* = 11	FEC/TE + I_2_ + FEC/TE + I_2_ *N* = 11	*P*-Value
1 grade neutropenia	7 (63.6%)	5 (41.6%)	5 (45.5%)	5 (45.5%)	0.72
2–3 grade neutropenia	0	0	4 (36.4%)	0	<0.003 *
1 grade nausea, vomiting and diarrhea	2 (18.1%)	1 (8.3%)	9 (81.2%)	3 (27.3%)	<0.002 *
2–3 grade nausea, vomiting and diarrhea	0	0	2 (18.1%)	0	0.090
Myalgia	8 (72.7%)	9 (75%)	9 (81.8%)	9 (75%)	0.99
Hand-foot syndrome	0	0	3 (27.3%)	0	<0.01 *

The National Cancer Institute Common Toxicity Criteria V4.0 (CTCAE; [18]). Values were obtained after the last chemotherapy FEC/TE for each group. * *P*-value by chi-square test.

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
