# Peer review of "Adjuvant Effect of Molecular Iodine in Conventional Chemotherapy for Breast Cancer. Randomized Pilot Study"

_nutrients, 2019, doi:10.3390/nu11071623_

Round 1
Reviewer 1 Report
In the submitted manuscript, Moreno-Vega et al. evaluate the effects of oral supplement for iodine alone and in combination with neoadjuvant chemotherapy on treatment response and on gene expression profiles.
Comments:
1. Lines 188 and 182, the authors wrote PD (progressive disease) when according to Figure 2B they are actually referring to PR (partial response) data.
2. What are the ER, PR, HER2 receptor status of the tumors of patients included in this study? The authors should include these variables to their analysis as they may find subgroups that benefit more from iodine supplementation.
3. Moreno-Vega et al. claim that iodine supplementation favors the re-induction of ESR1 expression in breast cancer. However, this could be due to clone selection, as iodine alone do not seem to stimulate ESR1 expression. The authors should comment on that.
Author Response
1. Lines 188 and 182, the authors wrote PD (progressive disease) when according to Figure 2B they are actually referring to PR (partial response) data.
The referee is right, this error it has been corrected.
2. What are the ER, PR, HER2 receptor status of the tumors of patients included in this study? The authors should include these variables to their analysis as they may find subgroups that benefit more from iodine supplementation.
We have the data of ER and PR, which are included in the complementary tables 2 and 3. Unfortunately, in 2005-2010, our Public Health System did not include the Her2 quantifications. However, we are aware that this marker is very important and can give us a more comprehensive view of the effect of iodine. In phase III of the clinical trial that is currently underway, we include all these parameters. Nevertheless, we have a paragraph in the discussion section that mentions this limitation (line 518).
3. Moreno-Vega et al. claim that iodine supplementation favors the re-induction of ESR1 expression in breast cancer. However, this could be due to clone selection, as iodine alone do not seem to stimulate ESR1 expression. The authors should comment on that.
The observation of the referee is relevant, indeed a clonal selection of estrogen-positive cells after treatment is also possible. We cannot rule out the participation of iodine in the re-differentiation in early tumors because in this work all were positive before and after treatment. We will include a paragraph with this second possibility.
Reviewer 2 Report
The work is innovative with a high scientific and clinical aspect. The results are clear and detailed, and the quality of the figures is very good. A brief introduction and discussion emphasize the high commitment of the Authors of the publication.
Author Response
The work is innovative with a high scientific and clinical aspect. The results are clear and detailed, and the quality of the figures is very good. A brief introduction and discussion emphasize the high commitment of the Authors of the publication.
We thank the referee for his valuable considerations towards our work.
Reviewer 3 Report
Manuscript titled “Adjuvant effect of molecular iodine in conventional chemotherapy for breast cancer. A randomised pilot study.” Through a series of elegantly designed experiments in this pilot study to address the hypothesis that molecular iodine prevents the detrimental side effects such as chemoresistance or cardiomyopathy of chemotherapy drugs or combination of drugs. The Authors developed further the observation of beneficial effects molecular iodine in breast cancer treatment from the animal studies to present study and now registered for phase III clinical trials.
An important observation that has medical and scientific merits and would contribute to breast cancer patients’ treatment and possibly applicable to other types of cancers treatment. The detailed introduction with relevant references; methods described adequately. Please expand on clinical details of inclusion criteria, on exclusion criteria, the exclusion of patients with “2) diagnosed with a concurrent severe and uncontrolled disease” requires further explanations. The last sentence in paragraph 2 of the method section needs rewriting to clarify why researchers, physicians and patients should know about the identification of treatment. In paragraph three of the methods, the authors should rewrite the Molecular iodine (I2) 5 mg/day for 170 days clearly and in more details. The consort chart, evaluation of the efficacy of treatment, follow up - described reasonably well. Result in the table format, and figures require a clear presentation with detailed legend and titles. Some of the writings on the figures are not readable. The statistical analysis correctly performed showed the statistical difference clearly. The Kaplan Meier analysis for disease-free survival and overall survival over fivers periods present the significant value of molecular iodine as an adjuvant to chemotherapy drugs. Various methods used to validate the results gives strong credibility to the study meticulously performed in these pilot studies. This manuscript is considerably longer with detailed presentations of results up to 10 figures and a comprehensive discussion that is extraordinarily long and need a reduction in size to avoid repetitions and redundancies. Authors should try to present a shorter version of the study, which does not require a presentation of all the genetic maps and detailed pathways analysis. The sample size is small, but it is acceptable for the pilot study, and in phase III trial, a larger cohort is necessary to obtain meaningful results. References are adequate; authors have used relevant references and the funding agencies for completion of this project. Overall, a manuscript showing significant finding: that in the larger cohort provide meaningful results that could help to prevent many undesirable side effects of the chemotherapeutic drugs. The immunomodulatory effect of molecular iodine can also contribute to the efficacy of immunotherapy.
Author Response
Manuscript titled “Adjuvant effect of molecular iodine in conventional chemotherapy for breast cancer. A randomised pilot study.” Through a series of elegantly designed experiments in this pilot study to address the hypothesis that molecular iodine prevents the detrimental side effects such as chemoresistance or cardiomyopathy of chemotherapy drugs or combination of drugs. The Authors developed further the observation of beneficial effects molecular iodine in breast cancer treatment from the animal studies to present study and now registered for phase III clinical trials.
An important observation that has medical and scientific merits and would contribute to breast cancer patients’ treatment and possibly applicable to other types of cancers treatment. The detailed introduction with relevant references; methods described adequately. Please expand on clinical details of inclusion criteria, on exclusion criteria, the exclusion of patients with “2) diagnosed with a concurrent severe and uncontrolled disease” requires further explanations.
The paragraph was rewritten
The last sentence in paragraph 2 of the method section needs rewriting to clarify why researchers, physicians and patients should know about the identification of treatment.
The importance of opening the files and that all the participants knew the group to which each patient belongs is that after the surgery all the patients would receive the iodine. This consideration for the patients was described in the informed consent. In the paragraph following this sentence, this situation is mentioned, which is why we believe it is clear., “After surgery, all patients received I2 supplement together with 4-6 cycles of a common chemotherapeutic cocktail …”
In paragraph three of the methods, the authors should rewrite the Molecular iodine (I2) 5 mg/day for 170 days clearly and in more details.
We complete the paragraph with: “that correspond to the maximum period that some patients required to complete their chemotherapy treatment”
The consort chart, evaluation of the efficacy of treatment, follow up - described reasonably well. Result in the table format, and figures require a clear presentation with detailed legend and titles. Some of the writings on the figures are not readable.
In the final version tables and figures will be with the maximal resolution.
This manuscript is considerably longer with detailed presentations of results up to 10 figures and a comprehensive discussion that is extraordinarily long and need a reduction in size to avoid repetitions and redundancies. Authors should try to present a shorter version of the study, which does not require a presentation of all the genetic maps and detailed pathways analysis.
We reorganized some of the results and sent figures 5 and 7 and table 2 to the complementary data.
We believe that the discussion is appropriate in length, since it is the first time that the effect of molecular iodine on human mammary cancer is described and its integral discussion is very important for us.
The sample size is small, but it is acceptable for the pilot study, and in phase III trial, a larger cohort is necessary to obtain meaningful results. References are adequate; authors have used relevant references and the funding agencies for completion of this project. Overall, a manuscript showing significant finding: that in the larger cohort provide meaningful results that could help to prevent many undesirable side effects of the chemotherapeutic drugs. The immunomodulatory effect of molecular iodine can also contribute to the efficacy of immunotherapy.
We appreciate your comments and pertinent suggestions to our work